# Intravenous vs. Oral Dose Comparison of Ibuprofen and Tramadol Combinations—Enantiomers, Metabolite, Linearity, and Sex-Related Effects: A Pharmacokinetics Randomized Clinical Trial

**DOI:** 10.3390/ph18030331

**Published:** 2025-02-26

**Authors:** Carmen Portolés-Díez, María Rosario Salas-Butrón, Ana Ascaso-del-Rio, Ana B. Rivas-Paterna, Leonor Laredo-Velasco, Carlos Calandria, Nuria Sanz, Annik Bergeron, Luis Santé, Emilio Vargas-Castrillón, Antonio Portolés-Pérez

**Affiliations:** 1Servicio de Anestesiología, Reanimación y Tratamiento del Dolor, Hospital Clínico San Carlos, 28040 Madrid, Spain; mariadelcarmen.portoles@salud.madrid.org (C.P.-D.); lsante@hotmail.com (L.S.); 2Instituto de Investigación Sanitaria del Hospital Clínico San Carlos, 28040 Madrid, Spain; mariadelrosario.salas@salud.madrid.org (M.R.S.-B.); ana.ascaso@salud.madrid.org (A.A.-d.-R.); ab.rivas@enf.ucm.es (A.B.R.-P.); leonor.laredo@salud.madrid.org (L.L.-V.); emilio.vargas@salud.madrid.org (E.V.-C.); 3Servicio de Farmacología Clínica, Hospital Clínico San Carlos, 28040 Madrid, Spain; 4Facultad de Enfermería, Universidad Complutense de Madrid, 28040 Madrid, Spain; 5Farmalider SA, 28108 Madrid, Spain; carloscalandria@farmalider.com (C.C.); nuriasanz@farmalider.com (N.S.); 6Altasciences, Laval, QC H7V 4B3, Canada; abergeron@altasciences.com; 7Facultad de Medicina, Universidad Complutense de Madrid, 28040 Madrid, Spain

**Keywords:** O-desmethyl-Tramadol, bioavailability, analgesics association, opioids, non-steroidal analgesics, clinical trial, opiates equipotent doses

## Abstract

**Background/Objectives:** Using a combination of analgesics allows for the use of lower doses of each, therefore, lowering risk of side effects. The study aims to estimate the bioavailability (pharmacokinetics of enantiomers and metabolites, as well as linearity and sex-related effects) of fixed doses combinations of Ibuprofen/Tramadol via an intravenous (IV) vs. oral route, and it is interesting to bridge the gap of equipotent doses by different routes. **Methods:** This was a randomized, open-label, crossover, five-period pharmacokinetics clinical trial, in which a single dose of each formulation [four different strengths of Ibuprofen 400 mg/Tramadol HCl (30, 31.5, 33, 37.5 mg), intravenous; Ibuprofen/Tramadol HCl 400 mg/37.5 mg, granules for oral solution], were administered to healthy volunteers. Enantiomers of Ibuprofen, of Tramadol, and of its main active metabolite O-desmethyl-Tramadol (M1) were measured, and pharmacokinetic parameters (maximal concentration (Cmax) and area under the concentration curve (AUC)) were estimated. Given the exploratory nature of the study, the sample size was small to provide sufficient power for comparisons of differences across all subgroups. The study was registered at Spanish register of clinical trials (REec), EudraCT code: 2017-001303-77. **Results:** Twelve subjects were recruited. Different patterns of rate and amount of the studied analytes are shown for oral and the several strengths of IV drugs tested. Ibuprofen, with an absolute oral bioavailability of 91%, showed an equivalent AUC of oral and IV administration. Tramadol showed an absolute oral bioavailability of 80%. **Conclusions:** Intravenous administration of Tramadol produces higher bioavailability (Cmax and AUClast) of the parent drug and lower of M1, than oral route. Dose normalized Cmax and AUClast of Tramadol and M1 were into the bioequivalence interval. Upon our pharmacokinetics study results, the intravenous dose of Tramadol should not be reduced when switching from oral dosing. No significant differences attributable to sex, once corrected by weight, were found.

## 1. Introduction

Acute pain is the most common type of pain [1]. For years, the need to adjust the dose of analgesics has been highlighted, given that the higher the analgesic dose, the greater the risk of side effects, sometimes without improving the desired effect. Any improvement in relief must be balanced against the increased risk of adverse effects [2]. Pain is multifactorial [3], so the combination of various analgesic agents that activate multiple pain-inhibitory pathways allow a synergistic effect to cover the required analgesia with a lower dose of each one (and, therefore, lowering the risk of side effects) [3,4,5,6]. Evidence shows that the combination of opioids and non-steroidal anti-inflammatory drugs (NSAIDs) improves efficacy and reduces the dose of individual drugs when compared with monotherapy [1,7,8,9,10].

Pharmacokinetics play an important role in identifying possible successful combinations as it is the first step to foresee how the different drug will be available and when, as well as consider if they have potential complementary pharmacodynamics [11]. Not all combinations lead to an enhanced analgesic effect or reduced adverse events, so each combination needs to be evaluated individually [6].

Opioid analgesics are therapeutically beneficial when used properly, and there is evidence that prescribing lower doses (i.e., emergency room or after surgery) is associated with lower rates of long-term use and possible opioid-use disorders [12]. NSAIDs are the opioid-sparing pillars of a multimodal therapy for pain [13]. One of the recommendations of the Enhanced Recovery After Surgery Society for the perioperative management of patients undergoing hip and knee replacement surgery includes a multimodal opioid-sparing analgesic regimen [14]. Studies indicate that the combination of NSAID with low doses of Tramadol show a significant reduction in pain scores [15] and provide superior analgesia than any of the components in monotherapy [16]. Some combinations of Tramadol with NSAID have been described and used effectively for decades (e.g., Tramadol/Dexketoprofen, Tramadol/Diclofenac, Tramadol/Celecoxib) and studied in diverse acute postoperative pain scenarios such as dental extraction, abdominal hysterectomy, or hip arthroplasty [16,17,18], although not with low doses of Tramadol, which is registered as a patent [19] and is not available for IV administration.

Ibuprofen is a nonsteroidal anti-inflammatory drug, a derivative of propionic acid with analgesic, antipyretic, and anti-inflammatory activity. It is rapidly absorbed, with Tmax being variable depending on the pharmaceutical formulation, and it is eliminated with an HL of about 2 h [20]. Its mechanism of action is based on the inhibition of cyclooxygenase, resulting in decreased prostaglandin synthesis. The pharmacological activity is largely attributed to its S-(+)-enantiomer, which is the active form inhibiting cyclooxygenase enzyme activity. Although the R-(−)-enantiomer is considered less active, it undergoes chiral inversion in vivo, converting partially to the active S-form (between 35 and 70%) [21], thus, contributing to its overall efficacy. Tramadol is a centrally acting analgesic, structurally related to codeine, and it is rapidly and almost completely absorbed and follows a first-pass effect of about 30% after oral administration, and the HL is about 6 h [22]. It consists of two enantiomers (dextro and levo enantiomers), both of which contribute to analgesic activity via different mechanisms [11,13]. The (+)-Tramadol enantiomer primarily acts as weak μ opioid receptor (MOR) agonist while the (−)-Tramadol enantiomer inhibits the reuptake of serotonin and norepinephrine, thereby enhancing the analgesic effect through modulation of these neurotransmitter pathways. This dual mechanism, involving both opioid receptor activation and monoamine reuptake inhibition, underscores the importance of enantiomer-specific activity in Tramadol’s overall analgesic profile [23]. As a member of the opioid family and given the current crisis of opioid abuse and misuse, it is crucial to adjust its dose.

The present study was designed to evaluate the pharmacokinetics characteristics of several strengths of low doses of Tramadol combined with Ibuprofen by intravenous (IV) vs. oral route in healthy volunteers. It allows to evaluate the absolute and comparative bioavailability of both drugs by different routes, the linearity in low level (previously not studied but the future of opiates saving strategies) doses of Tramadol, active metabolite and enantiomers, as well as secondarily analyze the effect of sex. Also, safety was assessed. All these characteristics are required for bridging the gap of the equipotential doses of Tramadol by intravenous vs. oral route for designing new medicines as well as for dosing optimization.

## 2. Results

### 2.1. Demographics

Twelve healthy subjects (seven male, five female) were recruited (8 to 29/05/2017), and there were no dropouts (follow-up: 28/05 to 03/07/2017). Figure 1 shows the flow chart of the study. Figure 1 shows the patient flow chart.

The population demographic characteristics of the study were as follows (mean ± SD): age (24.33 ± 5.25 year), weight (67.05 ± 11.30 kg; M: 74.7 ± 8.4, F: 56.4 ± 4.9), height (170.5 ± 8.99 cm), body mass index (22.89 ± 2.02 kg/m^2^).

### 2.2. Pharmacokinetics

The pharmacokinetics of Ibuprofen, Tramadol, and O-desmethyl-Tramadol were adequately characterized after a single dose of each formulation.

#### 2.2.1. Ibuprofen

Figure 2 and Figure 3 show R- and S- Ibuprofen enantiomer mean plasma concentration vs. time after oral and IV dosing, with very similar rate profiles.

Table 1 shows Ibuprofen -R and -S enantiomers pharmacokinetic parameters after oral and IV (T1 to T4) administration of 400 mg doses.

For both enantiomers, the comparison of the pharmacokinetic parameters obtained after oral (R) or IV (T1 to T4) administration of Ibuprofen 400 mg show overlapping 95%CI geometric mean intervals for AUClast and coincident Tmax medians, while Cmax is higher after IV dosing given the 95%CI did not overlap. The ANOVA for IV (T4) vs. oral (R) administration showed AUClast 90% CI into the 80–125% interval while being fully above it for Cmax (see Table 2). No difference is expected for the half-life (HL) after oral or IV administration.

As shown by the data, the absolute bioavailability of the oral studied formulation for Ibuprofen is 91%.

Table A1 (Appendix B) shows the pharmacokinetic parameters of Ibuprofen -R and -S by sex. The regression model showed a significant effect of the sex factor, not significant when weight was included in the model as an adjustment factor. No interaction between formulation and sex was observed. An example of the aforementioned regression model (Ibuprofen-S), including or not including weight as a confounding factor, is shown in Section A.1 (Appendix A).

#### 2.2.2. Tramadol

Figure 4 and Figure 5 show R- and S- Tramadol enantiomer mean plasma concentration vs. time.

Table 3 shows Tramadol -R and -S enantiomer pharmacokinetic parameters.

For both Tramadol enantiomers, the comparison of the pharmacokinetic parameters obtained after oral (R, Tramadol HCl 37.5 mg) or IV (T1 to T4, Tramadol HCl 30 to 37.5 mg) administration show overlapping 95%CI geometric mean intervals for AUClast from the different routes and doses, showing higher figures after IV than oral administration. The tendency of higher bioavailability after IV dosing is corroborated after normalizing the administered dose. Regarding Cmax, the IV administration leads to higher values, with non-overlapping 95%CI geometric mean intervals of any of the highest IV doses (T3 or T2 to T4) and the oral one (R). The difference is confirmed after adjusting by dose, with very similar Cmax/D values from the IV doses, non-overlapping any of them with the oral route. The ANOVA models confirm the higher bioavailability of Tramadol by intravenous vs. oral route (Table 2), and the extent of absorption after normalizing by the administered dose is into the 80–125% interval of equivalence (Table 4).

The median Tmax after all the IV formulations is 0.5 h, which is conditioned by the infusion duration, and a bit longer after oral administration (affected by absorption, transformation, and distribution processes). No difference is expected in HL after oral or IV administration.

From these data, the absolute bioavailability of Tramadol is calculated to be about 80%.

Table A2 (Appendix B) shows the pharmacokinetic parameters of Tramadol -R and -S by sex. The multiple regression model showed no significant effect of sex in the parameter (*p* > 0.05). No significant interaction between formulation and sex appeared.

#### 2.2.3. O-Desmethyl-Tramadol

Figure 6 and Figure 7 represent R- and S- O-desmethyl-Tramadol enantiomer mean plasma concentration vs. time. Figure A1 and Figure A2 (Appendix B) show the log-transformed concentrations vs. time of both enantiomers.

Table 5 shows O-desmethyl-Tramadol’s enantiomers pharmacokinetic parameters.

For both enantiomers, the comparison of the pharmacokinetic parameters obtained after oral (R, Tramadol HCl 37.5 mg) or IV (T1 to T4, Tramadol HCl 30 to 37.5 mg) administration show overlapping 95%CI geometric mean intervals for AUClast from the different routes and doses, showing higher figures after oral than IV administration.

The tendency of higher bioavailability after oral dosing is corroborated after adjusting by the administered dose. Regarding Cmax, the oral administration of Tramadol leads to higher values of its metabolite, especially the S-enantiomer which shows non-overlapping 95%CI geometric mean intervals of any of the IV doses (T1 to T4) and the oral one (R). The difference is confirmed after adjusting by dose, with very similar Cmax/D values from the IV doses, non-overlapping with the oral route. The ANOVA models show the lower availability (Cmax and AUClast) of O-desmethyl-Tramadol’s enantiomers by intravenous vs. oral administration (Table 2), and the extent of absorption after normalizing by the administered dose is into the 80–125% interval of equivalence (Table 4). The ANOVA test of differences between -S vs. -R enantiomers’ parameters, for the oral and intravenous administration, showed the following results (Ratio; 90% CI): Oral: Cmax: 159.82%; 110.94–230.25%; AUClast: 142.20%; 96.58–209.37%; Intravenous: Cmax: 148.01%; 106.51–205.65%, AUClast: 136%; 90.69–204.17%.

The median Tmax after the oral formulation is 1 h, with figures between 2 and 3.5 h after IV administration of the drug, indicating it takes longer to pass through the liver with respect to oral route. No difference is expected from the HL of the metabolite after oral or IV administration of the drug.

Table A3 (Appendix B) shows the pharmacokinetic parameters of O-desmethyl-Tramadol -R and -S by sex. The regression model showed a significant effect of the sex factor, which ceased to be significant when weight was included in the model as an adjustment factor. No interaction between formulation and sex was observed. An example of the aforementioned regression model (O-desmethyl-Tramadol-R), including or not including weight as a confounding factor, is shown in Section A.2.

### 2.3. Safety

Tolerability was considered good. A total of 19 adverse events were recorded, 16 of them with a possible or closer causal relationship to the study drugs (headache, 2; dizziness, 6; infusion site pain, 8). Three adverse events were recorded with a doubtful or not related causal relationship. All the adverse events were mild and transient. No severe events were recorded.

In the later safety analyses, some non-clinically significant changes were observed. Symptoms were not detected in any of the cases, and they were all considered as having no clinical significance.

## 3. Discussion

The present crisis of opioid abuse and misuse highlights the need of risk minimization and dose adjust more than ever. The current trend of drug combinations allows the use of lower doses of each drug and, therefore, reduces the risk of side effects with the same analgesic target [4,15]. The present study perfectly fits to the needs of information on new combinations characteristics, for the appropriate design of studies and use of medicines. Pharmacokinetics play a key role to identify possible successful combinations, based on their complementary pharmacokinetics and different mechanisms of action [11].

The advantages of fixed dose multimodal combinations in pain management are ease of administration, reduction in pill burden, potential for greater patient adherence, opioid sparing effect, dosing convenience, and greater analgesic efficacy with fewer adverse effects [13]. Some combinations of common analgesic and opioids have been studied such as Diclofenac/Tramadol, Dexketoprofen/Tramadol, Acetaminophen/Tramadol, and Acetaminophen/Codeine [1,2,4,8,15,17,18,24], although not yet with Ibuprofen plus low doses of Tramadol. Also, having IV formulations available is convenient when the oral route is not an option (e.g., in the operating room, post-surgery, or critical care). To our knowledge, this study is the first in assessing pharmacokinetics of oral and IV combinations of these drugs, enantiomers, metabolites, and sex effects.

The design and development of the study was appropriate to the aim and rules in force for pharmacokinetics (PK) studies. The wash out period was long enough to clear the drugs administered in the previous periods, as all the baseline analyses were below the limit of quantitation.

This study allowed a characterization of the pharmacokinetic profile of Ibuprofen and Tramadol, producing relevant information for the development and assessment of effects of these drugs and their combinations. Oral and IV administration of Tramadol show different patterns for the parent drug and metabolites due to their different absorption/distribution and metabolization processes. Of interest is the fact that one of the main metabolite of Tramadol, O-desmethyl-Tramadol metabolite (M1), also shows analgesic action by weak MOR agonism and has greater potency and affinity compared to the dextro form of Tramadol [18,25].

The bioavailablity (Cmax and AUClast) of Tramadol was evidenced as higher after intravenous than oral administration, while the contrary evidence was found for O-desmethyl-Tramadol, consistently with the O-demethylation of the drug in the liver, which produces higher transformation of oral Tramadol to M1. Both Tramadol and its main metabolite showed a linear relationship in the studied range of doses, being the 90% CI of the dose normalized Cmax and AUClast within the 80–125% interval of equivalence.

Both linearity and different pharmacokinetic processes, as well as the abovementioned higher potency of the metabolite with respect to the parent drug, are critical for the calculation of equipotent doses by different routes of administration or when switching to another opioid drug. The change in administration route should consider the relative potency of both the metabolite and the parent drug, while the change in dose can take advantage of a quite accurate estimation of the Cmax and AUClast obtained per mg of administered dose shown in the present study. Our findings provide clinically relevant insights into the optimization of Tramadol dosing, particularly in perioperative and acute pain management, where IV administration is often required. However, scientific evidence and dosage guidelines lack consensus and clarity regarding the conversion factor between oral and IV tramadol, with most sources failing to account for O-desmethyl-Tramadol concentrations despite its greater potency compared to tramadol. Surprisingly, the few sources recommending conversion factors suggest a dose reduction of around 20% when switching from oral to IV Tramadol, apparently based just on differences in tramadol bioavailability. This goes against the increase in dose apparently suggested by the lower concentration of the main active metabolite, M1, found after IV administration. Our study suggests that reducing the IV dose may not be appropriate, highlighting the need for individualized dosing strategies to optimize analgesic efficacy while minimizing adverse effects. Thus, it is unclear [19,25,26] whether higher (as stated by the lower concentrations shown by the active metabolite), lower [27,28], or similar [29] doses of IV tramadol, should be used when switching routes. Another adjustment option would be splitting in several doses administered IV every 2, 3, or 4 h, the total daily dose of Tramadol, as assessed by Lu et al. [27]. Upon our pharmacokinetics study results, the intravenous dose of Tramadol should not be reduced when switching from oral dosing. By providing a detailed pharmacokinetic profile of IV and oral Tramadol/Ibuprofen, our study contributes to the refinement of clinical guidelines and supports evidence-based decision making for dose adjustments. In addition, the infusion time can play a role to prevent excessive maximal concentrations; the study by Lu et al. [27] dosed Tramadol once at a time, while in our study it is administered over 30 min by an infusion pump, which is safer for the patient. At the end, PK modeling is of use in the design of new administration routes and dosing of medicines and the present study shows significant information for it. Turning now to the absolute oral bioavailability, the value observed for Tramadol in our study (81%) is consistent with that reported following 100 mg doses [25].

The -R and -S enantiomer pharmacokinetics (from both drugs Ibuprofen and Tramadol), showed similar pharmacokinetic behavior upon their 95%CI, while Cmax and AUClast of the enantiomers -R and -S of O-desmethyltramadol were not bioequivalent, reflecting stereoselective differences in their metabolism and distinct pharmacokinetic profiles, both in oral and intravenous administration.

Ibuprofen oral absorption is very fast, showing a similar Tmax in both oral and IV route, though Cmax is higher after IV administration. The absolute oral bioavailability found in our study for Ibuprofen is very high (91%), as expected, being AUClast into the 80–125% interval of equivalence when comparing intravenous/oral administration. Also, it has an early onset (30 min), not far from the IV route, given it was administered in 30′ infusion.

Regarding the sex effects, the differences suggested between male and female pharmacokinetic parameters (Cmax) for Ibuprofen and O-desmethyl-Tramadol enantiomers appeared only when body weight was drawn from the model, indicating that confusion was caused by this factor but not differences due to sex. Also, no significant interaction effect between the formulation (representing in this case the route of administration) and sex appeared. This information is relevant for clinical use of these drugs, and a pharmacokinetic perspective weight of the patient, not sex, should be considered for dosing.

Although not the main objective of the study, no concern was found regarding safety, which suggests the good tolerability of the combination.

As limitations of the study, it should be mentioned that the sample size was small and, although adequate statistical power was achieved for the primary analyses (comparisons achieved statistical power above 99%, with the exception of one comparison for tramadol, which was above 75%), it does not provide sufficient power for comparisons of differences across all subgroups. Also, the interval of doses of tramadol for evaluation of linearity may be considered narrow, although justified for the study of opiate low-dose combinations.

This line of research is of growing interest for the treatment of pain using opiates saving alternatives, so further drug combinations or dosages are expected to be evaluated. Of special interest for our formulation is the forthcoming demonstration of the analgesic effects of different strengths of the drug combination.

## 4. Materials and Methods

This is a randomized, open-label, crossover, five-period pharmacokinetics clinical trial, in which a single IV or oral dose of each combination was administered to healthy volunteers. The formulations were Ibuprofen (arginate)/Tramadol HCl IV in any of the four strengths (Test formulations: T1, T2, T3, T4) and Ibuprofen (arginine)/Tramadol HCl (Reference formulation: R, granules for oral solution). Enantiomers of Ibuprofen, of Tramadol, and of its main metabolite (M1) O-desmethyl-Tramadol were measured, and pharmacokinetic parameters were estimated. The comparative bioavailability of IV and oral routes was assessed, as well as linearity of Tramadol and its main metabolite through the interval of doses. According to guidelines on pharmacokinetics studies, 12 healthy volunteers were considered enough for the assessment of descriptive pharmacokinetics [30]. For the IV administration, a 400 mg Ibuprofen dose was selected (doses over 400 mg do not significantly increase analgesic effect) [31] combined with Tramadol HCl from 30 mg to 37.5 mg, lower than the standard recommended dose of 50 to 100 mg [26].

### 4.1. Subjects and Clinical Procedures

Individuals without relevant clinical alterations were included in the study; they freely accepted to participate in the study and signed the informed consent.

The study population was chosen from healthy volunteers of both sexes with the following inclusion criteria: male and female aged 18–40, body weight 50–100 kg, and body mass index 18.5–27 kg/m^2^, and no clinically/medically significant deviations from normal in the physical or electrocardiographic (twelve-lead electrocardiogram) examinations or in clinical laboratory tests (human immunodeficiency virus and viral hepatitis serology, hematology, biochemistry, urine, drugs of abuse, pregnancy test (women)).

A subject could not be selected for the study if they met any of the following exclusion criteria: smoking (within six weeks prior to study inclusion), pregnancy or lactation, history of drug abuse or alcoholism and/or use of any abused drugs within a month prior to the trial inclusion, stimulating drinks abuse (equivalent to 400 mg of caffeine per day), use of any treatment that could interfere with the study (enzyme inhibitors or inducers such as carbamazepine, erythromycin and phenytoin), participation in four clinical trials the previous year or in a clinical trial within 3 months prior to inclusion, significant medical condition or major surgery in the 3 previous months, impossibility to collaborate with the researchers, drug allergies, infectious disease such as HIV and hepatitis B/C, blood loss or blood donation (over 200 mL) within 3 months, disease that could alter drug’s absorption, distribution, metabolism or excretion, transfusion (blood or blood products) the previous 6 months, strenuous physical exercise within 72 h prior to admission, alcohol consumption within 48 h prior to admission, or history of peptic ulcer.

Sex perspective was considered, including healthy volunteers of both sexes, and including by-sex analysis and interactions. Given the kind of variables studied (pharmacokinetics), the gender perspective was not considered.

The study included five treatment periods, with a 7 day wash-out period between them. Given the number of periods (5) and the sample size, an incomplete randomization schema was applied, generated by computer (Program BSR 4.2) by the phase1 Unit, and the subjects assigned to the sequence in order of inclusion by the medical research team. The subjects were in-hospitalized in the Phase 1 studies the night previous to every drug administration. Starting at about 8.00 h. and with a few minutes of interval between volunteers, the medication was administered as a single dose in 10 h fasting conditions. During the study, water was allowed up to one hour before and two hours after drug administration. Food was allowed four hours after drug administration.

In order to determine the concentrations of the drugs, 15 blood samples (7 mL each, using dipotassium-Ethylenediaminetetraacetic acid as anticoagulant) per subject and period were drawn at the following times: baseline (pre-dose), +0.08, +0.25, +0.5, + 0.75, +1, +1.5, +2, +3, +4, +6, +8, +10, +12, +24 h post dose. The samples were centrifuged for 10 min at 1500× *g* and 4 °C, and the plasma was divided into four aliquots and frozen to −80 °C, then delivered frozen with appropriate inventory documentation and a data logger for monitoring the temperature during the transport to the bioanalytical center.

Laboratory analyses, clinical exams, and adverse events were recorded for the evaluation of safety.

### 4.2. Drugs

-Oral medication (Reference, R): Ibuprofen (arginine), corresponding to Ibuprofen 400 mg; Tramadol HCl 37.5 mg; granules for oral solution (R), orally administered with 200 mL of water.-Intravenous medication (Tests, T1 to T4): Ibuprofen (arginate), corresponding to Ibuprofen 400 mg/4 mL vials (manufactured by Braun laboratories) and Adolonta^®^ 100 mg/2 mL injectable solution were used. Immediately before its administration, it was extemporarily prepared on a 50 mL saline serum bottle, to reach the amount of Ibuprofen of 400 mg, and the amount of Tramadol HCl of 30 mg (T1), 31.5 mg (T2), 33 mg (T3), or 37.5 mg (T4), and infused in 30 min using an infusion pump.

All the drugs were provided by Farmalider SA, its administration and preparation procedure was standardized for all patients according to a specific drug administration guide.

### 4.3. Drug Analysis

Analytical methods were validated at and are proprietary to Altasciences, Laval, QC, Canada. The analyses were performed at blind, under Good Laboratory Practice rules, and Food and Drug Administration and European Medicines Agency guidelines compliance. Sample pre-treatment involved liquid–liquid extraction with methyl-tert-butyl-ether or chlorobutane using stable isotopic labeled internal standards. The enantiomers of Tramadol and O-desmethyl-Tramadol were quantified using a chiral column with hexane and ethanol mobile phase, followed by tandem mass spectrometry detection over theoretical concentration ranges of 1.00 ng/mL to 120.00 ng/mL for (+)-Tramadol and (-)-Tramadol, and 0.250 ng/mL to 75.000 ng/mL for (+)-O-desmethyl-Tramadol and (-)-O-desmethyl-Tramadol. The enantiomers of Ibuprofen were quantified using a chiral column with acidified methanol mobile phase, followed by MS/MS detection over a theoretical concentration range of 0.100 μg/mL to 50.000 μg/mL for both R-Ibuprofen and S-Ibuprofen.

The enantiomers of Tramadol and O-desmethyl-Tramadol were stable in human plasma for up to 120 days and Ibuprofen for up to 129 days when stored at −80 °C. All the analyses were performed at the end of the study, into the validated stability period.

The concentrations were calculated using peak area ratios, and the linearity of the calibration curve was determined using a weighted (1/x^2^) linear (y = mx + b) least squares regression analysis.

Method accuracy and reliability was demonstrated by between-run precision and accuracy. Between-run precision was 3.1% to 5.3% for Tramadol enantiomers, 1.8% to 3.5% for O-desmethyl-Tramadol, and 3.3% to 5.3% for Ibuprofen enantiomers. Between accuracy was 101.4% to 107.6% for Tramadol enantiomers, 99.5% to 104.4% for O-desmethyl-Tramadol enantiomers, and 97.4% to 108.6% for Ibuprofen enantiomers.

### 4.4. Pharmacokinetic Analyses

Upon the concentrations of each analyte (Ibu-R; Ibu-S; Tramadol-R; Tramadol-S; O-desmethyl-Tramadol-R; O-desmethyl-Tramadol-S), the pharmacokinetic parameters were estimated (using a non-compartmental method) and analyzed (descriptive statistics and Analysis of variance (ANOVA) without adjusting for multiple comparisons using NONLIN, Phoenix^®^, version 6.4.

The estimated parameters were as follows: Cmax (maximum plasma concentration); AUClast (area under the plasma concentration curve, calculated using the linear trapezoidal method until the last measurable concentration); Tmax (time until Cmax is reached); HL (plasma concentration half-life (0.693/b; b: Elimination constant calculated from the slope of the semilogarithmic representation of the concentration against time for the final part of the curve). In order to assess the linearity, Cmax/Dose and AUC/Dose were calculated for Tramadol and metabolites.

The pharmacokinetic parameters are evaluated descriptively, for the whole population and by-sex: mean, SD, 95% CI of mean (preferred for HL); GeoMean, 95% CI of GeoMean (preferred for Cmax and AUC); median, min, max (preferred for Tmax). Graphical descriptions were also used. Also, ANOVA models (including subject, formulation, period, and sequence) were used to evaluate bioequivalence between the intravenous vs. oral route of the same doses (formulation T4 vs. R) for Cmax and AUClast variables; to test differences between enantiomers of the M1 in both routes of administration (including subject, formulation, and period for each scenario), as well as to assess linearity between the highest vs. lowest intravenous doses (T4 vs. T1) for dose-normalized Cmax and AUClast. The usual bioequivalence criteria of 80.00–125.00% for the 90% CI of the ratios after Ln transformation is to be used as interval of reference.

To evaluate the potential effect of sex on the pharmacokinetic parameters Cmax and AUC for the same administered dose of Ibuprofen and Tramadol (formulations R and T4), multiple linear non-mixed effects regression analyses were conducted using Stata^®^ version 16.1. These analyses employed a backward stepwise method, incorporating sex (both alone and adjusted for body weight), formulation (reflecting the route of administration) and their first-order interactions. The potential confounder effect of body weight and sex is explained in detail in Appendix A. Initially, all potential predictors were included in the model. The backward stepwise method then removed predictors that did not significantly contribute to the model, based on their *p*-values (threshold of 0.05), until only significant variables remained. This approach ensures that the final model includes only the most relevant factors and provides a robust evaluation of the influence of sex, addressing body weight as a potential confounder in the interpretation.

### 4.5. Ethics

The protocol was authorized by the Spanish Agency of medicines and health products (EudraCT no.: 2017-001303-77, 25/04/2017, registered: REEC—Registro Español de Estudios Clínicos (aemps.es), https://reec.aemps.es/reec/public/web.html; accessed on 19 February 2025) and approved by the Ethics Committee of Instituto de Investigación Sanitaria Hospital Clínico San Carlos, Madrid, Spain (Internal code: 17/160-R_M, 21/04/2017). The study was conducted in accordance with the international ethical recommendations for investigation and clinical trials in humans contained in the Helsinki Declaration and its further revision as well as the recommendations on Good Clinical Practice (International Conference on Harmonization). The informed consent form was signed before each subject’s inclusion in the study, after receiving adequate information about the study design, objectives, and risks.

The data are available on reasonable request from the corresponding author due to confidentiality reasons.

## 5. Conclusions

Intravenous administration of Tramadol produces higher bioavailability (Cmax and AUClast) of the parent drug and lower for its main metabolite (M1), than oral route. Cmax and AUClast of Tramadol and M1 showed a linear increase with the dose after IV administration, and, when dose normalized, they were into the bioequivalence interval. Upon our pharmacokinetics study results, the intravenous dose of Tramadol should not be reduced when switching from oral dosing. No significant differences attributable to sex once corrected by weight were found.

## 6. Patents

Portolés A, Santé L, Salas M, Vargas E, Calandria C et al., inventors; Farmalider S.A., assignee. Combination of Ibuprofen and Tramadol for relieving pain, WO/2021/005129, 14 January 2021.

## Figures and Tables

**Figure 1 pharmaceuticals-18-00331-f001:**
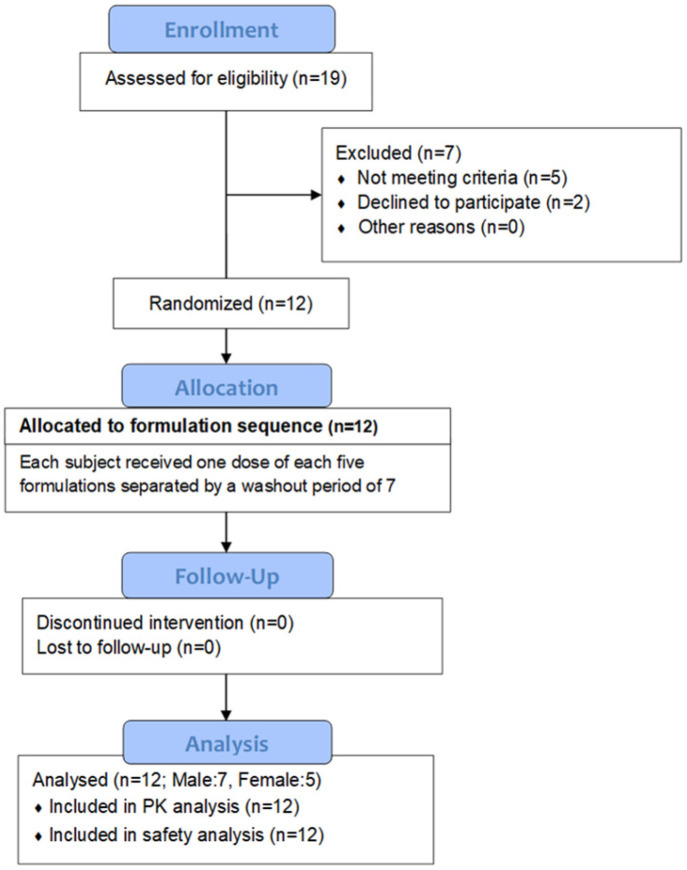
Flow Chart of patient disposition.

**Figure 2 pharmaceuticals-18-00331-f002:**
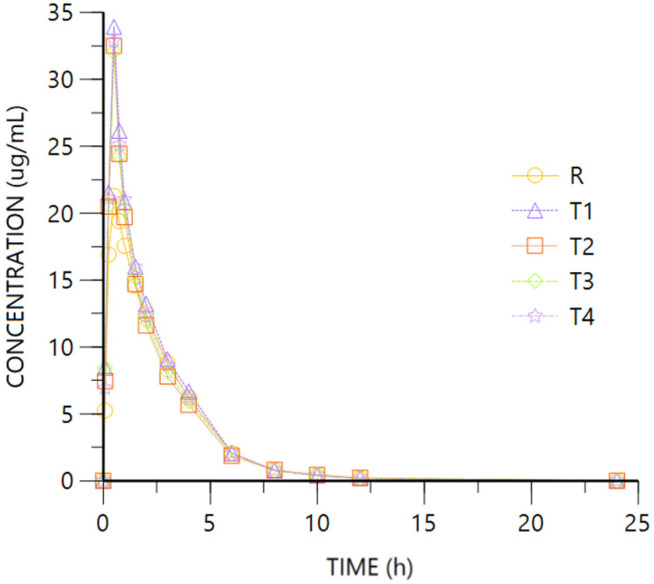
Ibuprofen R plasma concentration vs. time. R: Ibuprofen-Tramadol HCl 400–37.5 mg oral. T: Ibuprofen-Tramadol HCl (400-T1 30; T2 31.5; T3 33; T4 37.5 mg) IV.

**Figure 3 pharmaceuticals-18-00331-f003:**
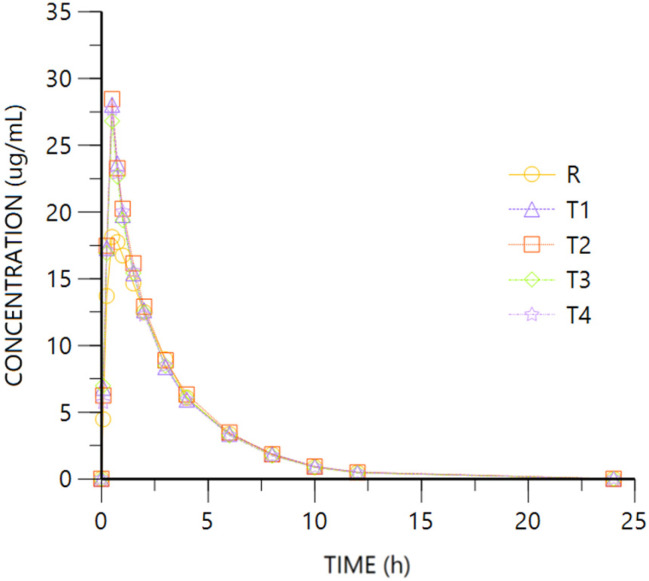
Ibuprofen S plasma concentration vs. time. R: Ibuprofen-Tramadol HCl 400–37.5 mg oral. T: Ibuprofen-Tramadol HCl(400-T1 30; T2 31.5; T3 33; T4 37.5 mg) IV.

**Figure 4 pharmaceuticals-18-00331-f004:**
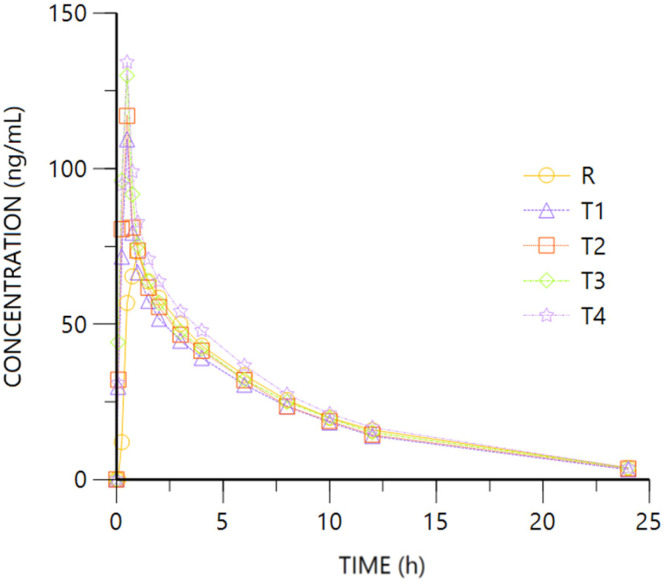
Tramadol R plasma concentration vs. time. R: Ibuprofen-Tramadol HCl 400–37.5 mg oral. T: Ibuprofen-Tramadol HCl (400-T1 30; T2 31.5; T3 33; T4 37.5 mg) IV.

**Figure 5 pharmaceuticals-18-00331-f005:**
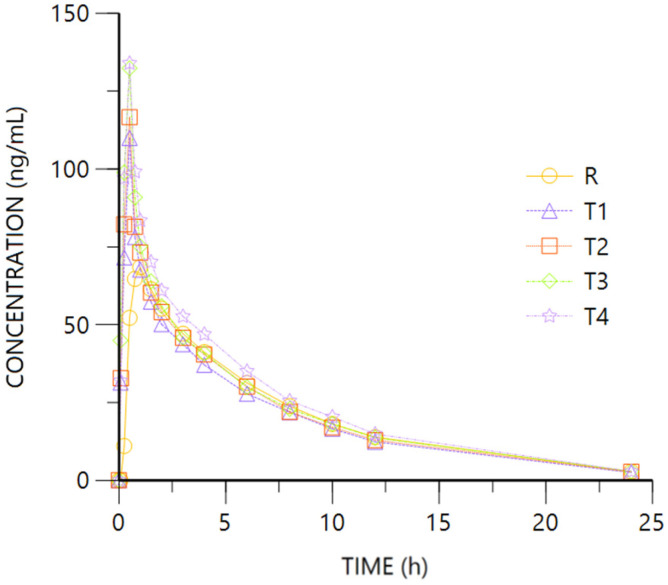
Tramadol S concentration vs. time. R: Ibuprofen-Tramadol HCl 400–37.5 mg oral. T: Ibuprofen-Tramadol HCl (400-T1 30; T2 31.5; T3 33; T4 37.5 mg) IV.

**Figure 6 pharmaceuticals-18-00331-f006:**
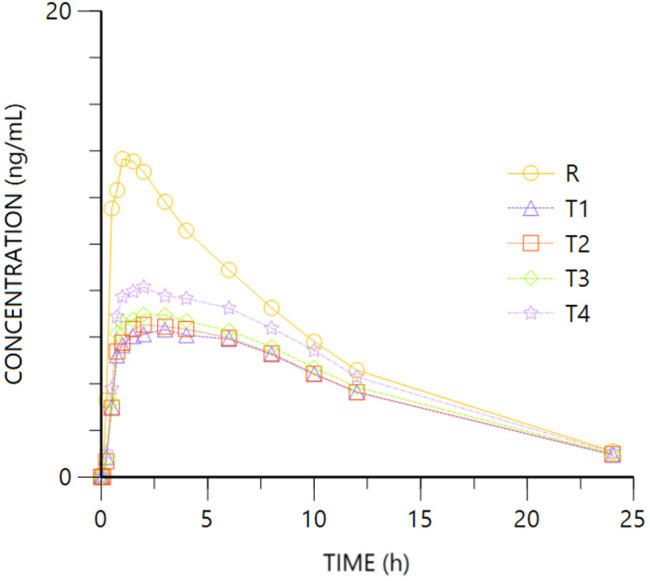
O-desmethyl-Tramadol R plasma concentration vs. time. R: Ibuprofen-Tramadol HCl 400–37.5 mg oral. T: Ibuprofen-Tramadol HCl (400-T1 30; T2 31.5; T3 33; T4 37.5 mg) IV.

**Figure 7 pharmaceuticals-18-00331-f007:**
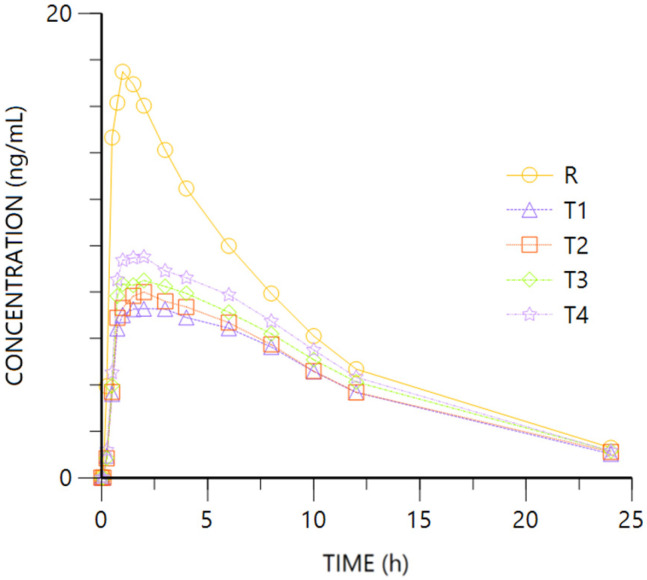
O-desmethyl-Tramadol S plasma concentration vs. time. R: Ibuprofen-Tramadol HCl 400–37.5 mg oral. T: Ibuprofen-Tramadol HCl (400-T1 30; T2 31.5; T3 33; T4 37.5 mg) IV.

**Table 1 pharmaceuticals-18-00331-t001:** Ibuprofen -R and -S pharmacokinetic parameters.

Variable (Units) Statistics	Formulation	Ibuprofen-R	Ibuprofen-S
**AUClast** (h·ug/mL) Geometric Mean [95%CI]	R	61.07 [53.18–70.14]	64.91 [57.25–73.61]
	T1	71.10 [64.06–78.92]	70.72 [61.73–81.00]
	T2	64.38 [56.22–73.71]	73.67 [67.04–80.96]
	T3	65.87 [55.99–77.50]	70.12 [63.67–77.21]
	T4	68.52 [60.46–77.65]	69.95 [62.28–78.56]
**Cmax** (ug/mL) Geometric Mean [95%CI]	R	22.42 [20.24–24.84]	19.34 [17.87–20.94]
	T1	33.66 [30.99–36.56]	27.84 [25.52–30.38]
	T2	32.03 [28.53–35.96]	28.11 [25.25–31.29]
	T3	31.60 [28.32–35.25]	26.56 [24.25–29.09]
	T4	32.64 [30.00–35.52]	27.46 [25.11–30.02]
**HL** (h) Mean (SD) [95%CI]	R	2.02 (1.71) [0.94–3.10]	2.06 (0.22) [1.93–2.20]
	T1	1.73 (0.84) [1.20–2.27]	2.12 (0.21) [1.99–2.26]
	T2	1.95 (0.73) [1.49–2.42]	2.31 (1.00) [1.68–2.95]
	T3	1.73 (0.62) [1.33–2.13]	2.06 (0.20) [1.93–2.19]
	T4	1.80 (0.57) [1.43–2.16]	2.04 (0.18) [1.93–2.15]
**Tmax** (h) Median [Min–Máx]	R	0.50 [0.25–1.00]	0.50 [0.25–1.00]
	T1	0.50 [0.50–0.50]	0.50 [0.50–0.75]
	T2	0.50 [0.50–0.50]	0.50 [0.50–0.50]
	T3	0.50 [0.50–0.50]	0.50 [0.50–0.50]
	T4	0.50 [0.50–0.50]	0.50 [0.50–0.50]

N: 12. R: Ibuprofen-Tramadol HCl 400–37.5 mg oral. T: Ibuprofen-Tramadol HCl (400-T1 30; T2 31.5; T3 33; T4 37.5 mg) IV.

**Table 2 pharmaceuticals-18-00331-t002:** ANOVA results on comparative intravenous/oral bioavailability of Ibuprofen (400 mg), Tramadol (37.5 mg), and metabolite enantiomers.

			Enantiomer-R			Enantiomer-S		
	Dependent	Units	Ratio_%	CI_90_ Lower	CI_90_ Upper	Ratio_%	CI_90_ Lower	CI_90_ Upper
Ibuprofen	Ln(Cmax)	ug/mL	146.54	137.03	156.71	141.41	133.82	149.43
	Ln(AUClast)	h·ug/mL	112.56	102.31	123.83	105.99	98.02	114.60
Tramadol	Ln(Cmax)	ng/mL	171.41	147.21	199.59	167.01	142.22	196.11
	Ln(AUClast)	h·ng/mL	123.23	113.18	134.17	128.08	117.56	139.55
O-desmethyl-Tramadol	Ln(Cmax)	ng/mL	60.64	54.98	66.89	55.89	51.02	61.22
	Ln(AUClast)	h·ng/mL	82.18	76.62	88.15	78.09	72.89	83.66

**Table 3 pharmaceuticals-18-00331-t003:** Tramadol -R and -S Pharmacokinetic parameters.

Variable (Units) Statistics	Formulation	Tramadol-R	Tramadol-S
**AUClast** (h·ng/mL) Geometric Mean [95%CI]	R	494.16 [370.42–659.24]	448.45 [331.87–605.99]
	T1	490.60 [391.40–614.94]	462.33 [372.32–574.10]
	T2	526.16 [436.58–634.10]	499.03 [416.80–597.49]
	T3	550.78 [456.61–664.37]	524.13 [438.76–626.12]
	T4	602.31 [492.16–737.12]	574.39 [472.16–698.75]
**AUClast/D** (h·ng/mL/mg) Geometric Mean [95%CI]	R	13.18 [9.88–17.58]	11.96 [8.85–16.16]
	T1	16.35 [13.05–20.50]	15.41 [12.41–19.14]
	T2	16.70 [13.86–20.13]	15.84 [13.23–18.97]
	T3	16.69 [13.84–20.13]	15.88 [13.30–18.97]
	T4	16.06 [13.12–19.66]	15.32 [12.59–18.63]
**Cmax** (ng/mL) Geometric Mean [95%CI]	R	80.29 [66.40–97.09]	76.53 [64.34–91.02]
	T1	105.76 [83.97–133.20]	105.60 [84.32–132.26]
	T2	112.12 [91.23–137.79]	112.79 [93.32–136.33]
	T3	124.37 [99.52–155.42]	126.70 [101.86–157.60]
	T4	128.72 [100.93–164.17]	127.81 [100.30–162.87]
**Cmax/D** (ng/mL/mg) Geometric Mean [95%CI]	R	2.14 [1.77–2.59]	2.04 [1.72–2.43]
	T1	3.53 [2.80–4.44]	3.52 [2.81–4.41]
	T2	3.56 [2.90–4.37]	3.58 [2.96–4.33]
	T3	3.77 [3.02–4.71]	3.84 [3.09–4.78]
	T4	3.43 [2.69–4.38]	3.41 [2.67–4.34]
**HL** (h) Mean (SD) [95%CI]	R	5.34 (1.48) [4.40–6.28]	4.86 (0.99) [4.23–5.49]
	T1	5.31 (1.38) [4.44–6.19]	5.05 (0.92) [4.46–5.63]
	T2	5.44 (1.47) [4.50–6.37]	5.02 (1.22) [4.25–5.79]
	T3	5.37 (1.55) [4.38–6.35]	4.91 (1.04) [4.25–5.57]
	T4	5.16 (1.48) [4.22–6.09]	4.70 (1.06) [4.03–5.37]
**Tmax** (h) Median [Min–Máx]	R	1.00 [0.50–2.00]	0.75 [0.50–2.00]
	T1	0.50 [0.25–0.75]	0.50 [0.25–0.75]
	T2	0.50 [0.50–1.00]	0.50 [0.50–1.00]
	T3	0.50 [0.50–1.00]	0.50 [0.50–0.75]
	T4	0.50 [0.25–0.75]	0.50 [0.25–0.75]

N: 12. R: Ibuprofen-Tramadol HCl 400–37.5 mg oral. T: Ibuprofen-Tramadol HCl (400-T1 30; T2 31.5; T3 33; T4 37.5 mg) IV.

**Table 4 pharmaceuticals-18-00331-t004:** ANOVA results on linearity of Tramadol and metabolite enantiomers between T4 vs. T1 intravenous doses.

			Enantiomer-R			Enantiomer-S		
	Dependent	Units	Ratio_%	CI_90_ Lower	CI_90_ Upper	Ratio_%	CI_90_ Lower	CI_90_ Upper
Tramadol	Ln(Cmax/D)	ng/mL/mg	100.73	86.39	117.45	99.35	84.84	116.34
	Ln(AUClast/D)	h·ng/mL/mg	96.96	88.98	105.66	97.76	89.36	106.95
O-desmethyl-Tramadol	Ln(Cmax/D)	ng/mL/mg	108.48	98.26	119.76	109.61	99.98	120.16
	Ln(AUClast/D)	h·ng/mL/mg	104.81	97.66	112.49	100.76	93.99	108.02

**Table 5 pharmaceuticals-18-00331-t005:** O-desmethyl-Tramadol -R and -S Pharmacokinetic parameters.

Variable (Units) Statistics	Formulation	O-Desmethyl-Tramadol-R	O-desmethyl-Tramadol-S
**AUClast** (h·ng/mL) Geometric Mean [95%CI]	R	104.66 [54.18–202.15]	148.82 [118.72–186.56]
	T1	64.54 [31.61–131.81]	90.52 [70.22–116.68]
	T2	67.18 [33.04–136.60]	95.45 [76.22–119.53]
	T3	72.70 [37.51–140.89]	104.41 [84.58–128.88]
	T4	82.84 [42.54–161.33]	112.73 [91.40–139.04]
**AUClast/D** (h·ng/mL/mg) Geometric Mean [95%CI]	R	2.79 [1.44–5.39]	3.97 [3.17–4.97]
	T1	2.15 [1.05–4.39]	3.02 [2.34–3.89]
	T2	2.13 [1.05–4.34]	3.03 [2.42–3.79]
	T3	2.20 [1.14–4.27]	3.16 [2.56–3.91]
	T4	2.21 [1.13–4.30]	3.01 [2.44–3.71]
**Cmax** (ng/mL) Geometric Mean [95%CI]	R	11.14 [5.39–23.04]	17.80 [12.75–24.86]
	T1	4.91 [2.52–9.58]	7.17 [5.36–9.59]
	T2	5.10 [2.56–10.17]	7.63 [5.57–10.47]
	T3	5.59 [2.94–10.63]	8.37 [6.25–11.21]
	T4	6.38 [3.29–12.37]	9.44 [7.10–12.56]
**Cmax/D** (ng/mL/mg) Geometric Mean [95%CI]	R	0.30 [0.14–0.61]	0.47 [0.34–0.66]
	T1	0.16 [0.08–0.32]	0.24 [0.18–0.32]
	T2	0.16 [0.08–0.32]	0.24 [0.18–0.33]
	T3	0.17 [0.09–0.32]	0.25 [0.19–0.34]
	T4	0.17 [0.09–0.33]	0.25 [0.19–0.34]
**HL** (h) Mean (SD) [95%CI]	R	7.23 (4.10) [4.62–9.84]	6.46 (1.71) [5.37–7.54]
	T1	7.27 (2.88) [5.44–9.10]	6.60 (1.32) [5.76–7.44]
	T2	9.69 (9.38) [3.73–15.65]	7.23 (2.06) [5.92–8.54]
	T3	6.63 (2.01) [5.28–7.98]	6.91 (2.14) [5.55–8.27]
	T4	7.98 (7.01) [3.28–12.69]	6.61 (1.82) [5.45–7.77]
**Tmax** (h) Median [Min–Máx]	R	1.00 [0.50–6.00]	1.00 [0.50–2.00]
	T1	3.00 [1.00–8.00]	2.50 [0.75–6.00]
	T2	3.50 [0.75–8.00]	2.00 [0.75–6.00]
	T3	2.00 [0.75–8.00]	1.50 [0.75–6.00]
	T4	2.00 [0.75–10.00]	2.00 [0.75–4.00]

N: 12. R: Ibuprofen-Tramadol HCl 400–37.5 mg oral. T: Ibuprofen-Tramadol HCl (400-T1 30; T2 31.5; T3 33; T4 37.5 mg) IV.

## Data Availability

The data are available on reasonable request from the corresponding author due to confidentiality reasons.

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
