# Peer review of "Intravenous vs. Oral Dose Comparison of Ibuprofen and Tramadol Combinations—Enantiomers, Metabolite, Linearity, and Sex-Related Effects: A Pharmacokinetics Randomized Clinical Trial"

_pharmaceuticals, 2025, doi:10.3390/ph18030331_

Round 1

Reviewer 1 Report

Comments and Suggestions for Authors

The introduction covers relevant background on multimodal analgesia and the pharmacokinetics of tramadol and ibuprofen. However, it could be strengthened by providing more recent literature and a more comprehensive discussion of the rationale for choosing these specific doses and formulations.

The methods are generally well detailed, but additional clarification on the statistical power calculations, potential confounders, and analytical validation could enhance reproducibility.

In addition, the statistical analysis section should explicitly mention whether corrections for multiple comparisons were applied.

The discussion on the bioavailability and metabolite formation is well-articulated. However, further elaboration on how these findings translate into clinical implications would strengthen the manuscript.

The manuscript’s readability could be improved by simplifying complex sentences and ensuring consistency in terminology

The manuscript is generally well-written, but minor grammatical and syntactic errors should be corrected. A professional language edit would enhance clarity.

Author Response

Thanks for taking your time to revise the manuscript. I’m firmly convinced that it has improved significantly thanks to the corrections. Please find below the detailed responses to your comments/questions.  

COMMENT 1: The introduction covers relevant background on multimodal analgesia and the pharmacokinetics of tramadol and ibuprofen. However, it could be strengthened by providing more recent literature and a more comprehensive discussion of the rationale for choosing these specific doses and formulations.

RESPONSE:

Thanks for your insightful comment. Attending to it, the introduction and discussion have been revised, some paragraphs reworded, and the references revised. At the end, seventeen references are dated in the last 4 years.

COMMENT  2: The methods are generally well detailed, but additional clarification on the statistical power calculations (1), potential confounders (2), and analytical validation (3) could enhance reproducibility.

RESPONSE:

Thank you for your comments. In response to your request, we provide additional details below:

  1. Regarding statistical power, the manuscript states that adequate statistical power was achieved for the primary analyses. This was calculated using NONLIN, Phoenix® version 6.4 as the power to detect a difference in least square means equal to 20% of the reference LSM. With respect to regression, power adhoc was calculated based on Cohen's effect size for multiple regression, using f2=R2/(1−R2), with G*Power.

Attending to your comment, we have now specified in the Discussion section what follows:

… comparisons achieved statistical power above 99%, with the exception of one specific comparison for tramadol, which had a power above 75%.

  1. Regarding potential confounders, as stated in the Materials and Methods section of the manuscript, the ANOVA included Formulation (Route), Sequence and Period, as well as subject(sequence) (the latter as random effect). For the Multiple Regression models, the text specifies (Material and Methods section) that the analysis followed a backward stepwise method, incorporating the potential confounders sex, weight, and formulation (reflecting the route of administration) and their first-order interactions to account for potential confounding effects. Also Appendix A shows the analyses of the terms, including interactions and confounding factors.

Attending to your comment, the explanation has been completed adding the following phrase in the Analyses subsection of Material and methods section:

The potential confounder effect of body weight and sex is explained in detail in Appendix A.

  1. Regarding analytical validation, the manuscript provides:

…the analytical methods were validated under GLP guidelines and in compliance with FDA and EMA regulatory standards, with analyses conducted in a blinded manner. Sample preparation involved liquid-liquid extraction with methyl-tert-butyl-ether or chlorobutane, using stable isotopically labeled internal standards. Chromatographic separation and detection were performed using a chiral column with hexane/ethanol mobile phase and tandem mass spectrometry (MS/MS) for Tramadol and O-desmethyl-Tramadol enantiomers, while Ibuprofen enantiomers were analyzed using a chiral column with acidified methanol mobile phase and MS/MS detection. Calibration ranges are also included: 1.00–120.00 ng/mL for Tramadol enantiomers, 0.250–75.000 ng/mL for O-desmethyl-Tramadol enantiomers, and 0.100–50.000 μg/mL for Ibuprofen enantiomers. Stability assessments confirm that Tramadol and O-desmethyl-Tramadol enantiomers remained stable for 120 days, and Ibuprofen enantiomers for 129 days at -80°C. Precision (between-run) is also specified (3.1–5.3% for Tramadol enantiomers, 1.8–3.5% for O-desmethyl-Tramadol enantiomers, and 3.3–5.3% for Ibuprofen enantiomers, while accuracy (between-run) was 101.4–107.6% for Tramadol enantiomers, 99.5–104.4% for O-desmethyl-Tramadol enantiomers, and 97.4–108.6% for Ibuprofen enantiomers). Regarding data processing, we specify that concentrations were calculated using peak area ratios, and calibration curve linearity was assessed via weighted (1/x²) least squares regression.

The laboratory (Altasciences, Canada) has proprietary and validated methods, protected by confidentiality reasons, thus they are not allowed to give more detailed information.

The laboratory contact information is included in the manuscript, and any person interested in the method can contact them as convenient.

In terms of reliability, the laboratory is among the more experienced in the world, working under GLP rules and compliant with FDA and EMA guidelines, and they explained as follows:

as per bioanalytical reports, the sample analysis of the FLR-P3-350 study was conducted in accordance with FDA Guidance for the Industry, Bioanalytical Method Validation (May 2001) and EMA Guideline on Bioanalytical Method Validation (EMEA/CHMP/EWP/192217/2009 Rev.1 Corr. 2**). 

The study was conducted, and the data was generated, documented, reviewed and reported:

  • in accordance with Good Clinical Practices (GCPs), as described by ICH E6 GCP;
  • in accordance with the applicable principles of Good Laboratory Practices (GLPs), as described by FDA Title 21 CFR Part 58 and by OECD Principles of Good Laboratory Practice, ENV/MC/CHEM(98)17 (as revised in 1997);
  • in accordance with the Reflection paper for laboratories that perform the analysis or evaluation of clinical trial samples (EMA/INS/GCP/532137/2010);
  • in accordance with the research protocol, the Bioanalytical Plan, and applicable SOPs.

Also signficant is the fact that the level of information supplied has been considered enough for publishing in every other case.

COMMENT 3: In addition, the statistical analysis section should explicitly mention whether corrections for multiple comparisons were applied.

RESPONSE:

Thank you for your comment. Given the exploratory nature of the study and its primary focus on pharmacokinetic parameter estimation rather than hypothesis testing across multiple independent endpoints, the analyses were conducted without adjusting for multiple comparisons. Instead, we followed standard pharmacokinetic study guidelines, where the interpretation of results is based on confidence intervals and predefined bioequivalence criteria rather than significance testing across multiple outcomes. We believe this approach is appropriate given the study design and objectives.

Thus, attending to your comment, we have clarified that no corrections for multiple comparisons were applied in the statistical analysis, including in the Analyses section the following phrase:

… without adjusting for multiple comparisons…

COMMENT  4: The discussion on the bioavailability and metabolite formation is well-articulated. However, further elaboration on how these findings translate into clinical implications would strengthen the manuscript.

RESPONSE:

Thank you for your valuable feedback. We have expanded the discussion to better highlight the clinical implications of our findings. Specifically, we have emphasized that the higher bioavailability of IV Tramadol suggests that dose reduction when switching from oral to IV may not be necessary, contrary to some existing recommendations. That our pharmacokinetic data provide important insights for optimizing Tramadol dosing, particularly where IV administration is often necessary. That individualized dosing strategies should be considered to maintain analgesic efficacy while minimizing adverse effects. And that our findings reinforce the need for evidence-based dose adjustments and contribute to refining clinical guidelines for Tramadol/Ibuprofen use.

The following phrases have been added to the discussion:

. Our findings provide clinically relevant insights into the optimization of Tramadol dosing, particularly in perioperative and acute pain management, where IV administration is often required.

. Our study suggests that reducing the IV dose may not be appropriate, highlighting the need for individualized dosing strategies to optimize analgesic efficacy while minimizing adverse effects.

. … should be used when switching routes, or whether doses or may need to…

. By providing a detailed pharmacokinetic profile of IV and oral Tramadol/Ibuprofen, our study contributes to the refinement of clinical guidelines and supports evidence-based decision-making for dose adjustments.

COMMENT  5: The manuscript’s readability could be improved by simplifying complex sentences and ensuring consistency in terminology

RESPONSE:

Thank you for pointing this out. We have revised the entire document trying to avoid complex sentences and improve consistence in terminology and abbreviations.

COMMENT 6: The manuscript is generally well-written, but minor grammatical and syntactic errors should be corrected. A professional language edit would enhance clarity.

RESPONSE:

Thanks for your comment, a complete revision has been made to avoid syntactic errors.

Reviewer 2 Report

Comments and Suggestions for Authors

This is an interesting cross-over clinical trial that compared the pharmacokinetics of IV and oral tramadol. I only have minor comments:

1. The title is too long and a little confusing, could you make it more concise?

2. In Methods, "clinically/medically significant deviations from normal in the physical or electrocardiographic examinations or in clinical laboratory tests", please provide more details for electrocardiographic examinations and clinical laboratory tests.

Author Response

Thanks for taking your time to revise the manuscript. I’m firmly convinced that it has improved significantly thanks to the corrections. Please find below the detailed responses to your comments/questions.  

This is an interesting cross-over clinical trial that compared the pharmacokinetics of IV and oral tramadol. I only have minor comments:

COMMENT 1: The title is too long and a little confusing, could you make it more concise?

RESPONSE:

We agree, attending to your comment the title has been modified as follows:

Intravenous vs oral dose comparison of Ibuprofen and Tramadol Combinations. Enantiomers, metabolite, linearity, and sex-related effects, pharmacokinetics randomized clinical trial

COMMENT 2: In Methods, "clinically/medically significant deviations from normal in the physical or electrocardiographic examinations or in clinical laboratory tests", please provide more details for electrocardiographic examinations and clinical laboratory tests.

RESPONSE:

We agree, the corresponding paragraph has been modified as follows:

…no clinically/medically significant deviations from normal in the physical or electrocardio-graphic (twelve-lead electrocardiogram) examinations or in clinical laboratory tests (Human immunodeficiency virus and viral hepatitis serology, hematology, biochemistry, urine, drugs of abuse, pregnancy test –women-).

Reviewer 3 Report

Comments and Suggestions for Authors

The study is well-designed and addresses an important clinical question [Ibuprofen and Tramadol Fixed-Dose Combinations and pharmacokinetics data]. The results are presented clearly and support the conclusions. However, many revisions are required to improve clarity.

1-     In the abstract, Ibuprofen, with an absolute oral bioavailability of 91%, showed equivalent AUC of oral and IV administration. What about tramadol bioavailability ??

2-     In the abstract, it would be helpful to include a statement on the clinical implications of the findings, particularly how the results might influence dosing strategies in practice.

3-     In the abstract, ensure that all abbreviations (e.g., AUC, Cmax) are defined upon first use.

4-     In the abstract, consider adding a brief statement on the limitations of the study, if any, to provide a balanced view.

5-     For keywords add Ibuprofen

6-     The research gap should be highlighted more in the abstract and the introduction ; are they any related studies for effect of tramadol on pharmacokinetics of NSAIDs ?

7-     Pharmacokinetics data for both drugs should be reported in the introduction

Vazzana, M., Andreani, T., Fangueiro, J., Faggio, C., Silva, C., Santini, A., Garcia, M.L., Silva, A.M. and Souto, E.B., 2015. Tramadol hydrochloride: pharmacokinetics, pharmacodynamics, adverse side effects, co-administration of drugs and new drug delivery systems. Biomedicine & Pharmacotherapy70, pp.234-238.

Jin, Y., Zhang, M., Di, X., Qi, X., Zheng, L. and Wang, Z., 2023. Comparison of intravenous ibuprofen pharmacokinetics between Caucasian and Chinese populations using physiologically based pharmacokinetics modeling and simulation. European Journal of Pharmaceutical Sciences191, p.106587.

8-     In line 97, write full name for BMI

9-     In line 128, reasons for this finding should be stated [The figures for Cmax were found to be higher for women than men]. Also. In line 166.

10-  In line 158, write full name for HL

11-  In line 280, write units for 30.

12-  In the paragraph 297-304, regarding the sex effect; alignment should be reported between this paragraph and what mentioned before in lines 128, 166 where The figures for Cmax were found to be higher for women than men

13-  The paragraph from line 305 to line 310; should be shifted to the introduction section with further details about mutual effects on the pharmacokinetic.

14-  In line 330, write the nationality of  participants in this study , the genetic factor may affect results.

15-  In line 384, write full names for all abbreviations when first mentioning [under GLP rules, FDA and EMA guidelines]

16-  Add reference for drug analytical methods

17-  In ethical approval section [The protocol was authorized by the AEMPS (EudraCT no.: 2017-001303-77) and approved by the Ethics Commi􀁅ee of Instituto de Investigación Sanitaria Hospital Clínico  San Carlos (Internal code: 17/160-R_M).  write full names for abbreviations ( AEMPS) and town country for the hospital.

18-  For funding and acknowledgement  statements add town and country

19-  All mentioned abbreviations should be included in abbreviation list; for example FDA,GLP] , please read the text well and include all abbreviations

20-  Generally, ensure that all abbreviations in the text are defined upon first use.

21-  Study limitations and future plan should be stated in more details

Author Response

Thanks for taking your time to revise the manuscript. I’m firmly convinced that it has improved significantly thanks to the corrections. Please find below the detailed responses to your comments/questions.  

The study is well-designed and addresses an important clinical question [Ibuprofen and Tramadol Fixed-Dose Combinations and pharmacokinetics data]. The results are presented clearly and support the conclusions. However, many revisions are required to improve clarity.

COMMENT 1:     In the abstract, Ibuprofen, with an absolute oral bioavailability of 91%, showed equivalent AUC of oral and IV administration. What about tramadol bioavailability ??

RESPONSE:

Thanks for your comment. We agree that it is expected to find the corresponding absolute oral bioavailability parameter for Tramadol, in parallel with Ibuprofen.

Thus, the following phrase has been included in the abstract:

Tramadol, showed an absolute oral bioavailability of 80%.

COMMENT 2:     In the abstract, it would be helpful to include a statement on the clinical implications of the findings, particularly how the results might influence dosing strategies in practice.

RESPONSE:

Thanks for your comment, we agree that information could be of interest in the abstract, accordingly the following sentence has been included in the conclusions subsection:

Upon our pharmacokinetics study results, the intravenous dose of Tramadol should not be reduced when switching from oral dosing.

COMMENT 3:     In the abstract, ensure that all abbreviations (e.g., AUC, Cmax) are defined upon first use.

RESPONSE:

Although the abbreviations are mentioned in the corresponding section at the end of the document, attending to your comment they have also been defined after its first use, in the abstract or along the manuscript.

COMMENT 4:     In the abstract, consider adding a brief statement on the limitations of the study, if any, to provide a balanced view.

RESPONSE:

Thanks for yous insightful comment. Accordingly, we have included in the abstract the following phrase:

Given the exploratory nature of the study, the sample size was small to provide sufficient power for comparisons of differences across all subgroups.

COMMENT 5:     For keywords add Ibuprofen

RESPONSE:

We agree that it could be expected to find Ibuprofen as a keyword, but as far as it is mentioned in the Title, it is not needed to repeat it again, as the terms in the title are by themselves used as search ítems.

COMMENT 6:    a) The research gap should be highlighted more in the abstract and the introduction ; b) are they any related studies for effect of tramadol on pharmacokinetics of NSAIDs ?

RESPONSE:

a) Attending to your comment, the text has been modified as follows:

Abstract:

… The study aims to estimate the bioavailability (pharmacokinetics of enantiomers and metabolites, as well as linearity and sex-effects) of fixed doses combinations of Ibuprofen/Tramadol intravenous (IV) vs oral route, interesting to bridge the gap of equipotent doses by different routes. …

Introduction (two last paragraphs have been reworded as follows):

The present study was designed to evaluate the pharmacokinetics characteristics of several strengths of low doses of Tramadol combined with Ibuprofen by intravenous (IV) vs oral route in healthy volunteers. It allows to evaluate the absolute and comparative bioavailability of both drugs by different routes, the linearity in low level (previously not studied but the future of opiates saving strategies) doses of Tramadol, active metabolite and enantiomers, as well as secondarily analyze the effect of sex. Also, safety was assessed. All these characteristics are required for bridging the gap of the equipotential doses of Tramadol by intravenous vs oral route for designing new medicines as well as for dosing optimization.

b) Regarding your question about other studies on the effect of tramadol on pharmacokinetics of NSAIDs, it is important to note that the study does not intend to test such effect (as it would require comparison of the drugs alone vs combination), but the characteristics of the fixed-dose combinations by different routes.

COMMENT 7:     Pharmacokinetics data for both drugs should be reported in the introduction

Vazzana, M., Andreani, T., Fangueiro, J., Faggio, C., Silva, C., Santini, A., Garcia, M.L., Silva, A.M. and Souto, E.B., 2015. Tramadol hydrochloride: pharmacokinetics, pharmacodynamics, adverse side effects, co-administration of drugs and new drug delivery systems. Biomedicine & Pharmacotherapy70, pp.234-238.

Jin, Y., Zhang, M., Di, X., Qi, X., Zheng, L. and Wang, Z., 2023. Comparison of intravenous ibuprofen pharmacokinetics between Caucasian and Chinese populations using physiologically based pharmacokinetics modeling and simulation. European Journal of Pharmaceutical Sciences191, p.106587.

RESPONSE:

Attending to your comment, the PK data of both drugs have been mentioned in the Introduction.

COMMENT 8:     In line 97, write full name for BMI

RESPONSE:

Thanks for your comment. We agree that, given it is mentioned only once, the acronym  is not needed and the full name can be used.

COMMENT 9:     In line 128, reasons for this finding should be stated [The figures for Cmax were found to be higher for women than men]. Also. In line 166.

RESPONSE:

Thanks for your insightful comment, after revising the statements and writing we decided to avoid those comments on the descriptive exploration of data on the maximal concentrations of every drug.

COMMENT 10:  In line 158, write full name for HL

RESPONSE:

We agree, the full name has been included.

COMMENT 11:  In line 280, write units for 30.

RESPONSE:

We agree, the units (minutes) have been included instead of the notation ‘

COMMENT 12:  In the paragraph 297-304, regarding the sex effect; alignment should be reported between this paragraph and what mentioned before in lines 128, 166 where The figures for Cmax were found to be higher for women than men

RESPONSE:

Please see answer to question 9 above. After erasing the descriptive comments on the concentration, we consider the paragraph in the discussion clear enough.

COMMENT 13:  The paragraph from line 305 to line 310; should be shifted to the introduction section with further details about mutual effects on the pharmacokinetic.

RESPONSE:

Attending to your comment the paragraph has been moved to the Introduction section, and the references adapted accordingly.

COMMENT 14:  In line 330, write the nationality of  participants in this study , the genetic factor may affect results.

RESPONSE:

Thanks for your insightful comment, we agree that genetic factors could affect pharmacokinetics, although it is not expected to impact our study. In this case, 11 subjects were caucasic and 1 asiatic, the latter didn’t show apparent differences to the rest. Moreover, being a cross-over study, only the variability could be affected, as any factor present in the subjects would be controlled by the same sample of subjects in every group. For this reason, the analysis of race effects were out of the scope of our study.

COMMENT 15:  In line 384, write full names for all abbreviations when first mentioning [under GLP rules, FDA and EMA guidelines]

RESPONSE:

Agree, the full names of the mentioned abbreviations have been included

COMMENT 16:  Add reference for drug analytical methods

RESPONSE:

Thanks for your insightful comment. After discussing this matter with the laboratory in charge of the drug analyses, we consider the level of detail enough for the aim of the publication, upon the following reasons:

The laboratory (Altasciences, Canada) has proprietary and validated methods, protected by confidentiality reasons, thus they are not allowed to give more detailed information.

The laboratory contact information is included in the manuscript, and any person interested in the method can contact them as convenient.

In terms of reliability, the laboratory is among the more experienced in the world, working under GLP rules and compliant with FDA and EMA guidelines, and they explained as follows:

as per bioanalytical reports, the sample analysis of the FLR-P3-350 study was conducted in accordance with FDA Guidance for the Industry, Bioanalytical Method Validation (May 2001) and EMA Guideline on Bioanalytical Method Validation (EMEA/CHMP/EWP/192217/2009 Rev.1 Corr. 2**). 

The study was conducted, and the data was generated, documented, reviewed and reported:

  • in accordance with Good Clinical Practices (GCPs), as described by ICH E6 GCP;
  • in accordance with the applicable principles of Good Laboratory Practices (GLPs), as described by FDA Title 21 CFR Part 58 and by OECD Principles of Good Laboratory Practice, ENV/MC/CHEM(98)17 (as revised in 1997);
  • in accordance with the Reflection paper for laboratories that perform the analysis or evaluation of clinical trial samples (EMA/INS/GCP/532137/2010);
  • in accordance with the research protocol, the Bioanalytical Plan, and applicable SOPs.

Also signficant is the fact that the level of information supplied has been considered enough for publishing in every other case.

COMMENT 17:  In ethical approval section [The protocol was authorized by the AEMPS (EudraCT no.: 2017-001303-77) and approved by the Ethics Commi?ee of Instituto de Investigación Sanitaria Hospital Clínico  San Carlos (Internal code: 17/160-R_M).  write full names for abbreviations ( AEMPS) and town country for the hospital.

RESPONSE:

Agree, the full name and city of the hospital have been included.

COMMENT 18:  For funding and acknowledgement  statements add town and country

RESPONSE:

Agree, the city and country has been added as required.

COMMENT 19:  All mentioned abbreviations should be included in abbreviation list; for example FDA,GLP] , please read the text well and include all abbreviations

RESPONSE:

Thanks for your comment. In accordance to it, the full document has been revised and the abbreviation appropiately addressed.

COMMENT 20:  Generally, ensure that all abbreviations in the text are defined upon first use.

RESPONSE:

Thank you for pointing this out. Accordingly, the full document has been revised and the abbreviation appropiately addressed.

COMMENT 21:  Study limitations and future plan should be stated in more details

RESPONSE:

Attending to your insightful comment, we have added to the discussion the following paragraphs:

Also, the interval of doses of tramadol for evaluation of linearity may be considered narrow, although justified for the study of opiates low-dose combinations.

This line of research is of growing interest for the treatment of pain using opiates saving alternatives, so further drug combinations or dosages are expected to be evaluated. Of special interest for our formulation is the forthcoming demonstration of the analgesic effects of different strengths of the drug combination.

Round 2

Reviewer 3 Report

Comments and Suggestions for Authors

The authors did all required recommendations. I appreciate their responses. The paper could be published in the current form.